# Cosmologically-Coupled Black Holes and Dark Energy: Alleviating the Hubble Tension

## Abstract

This paper investigates the hypothesis that cosmologically-coupled black holes (CCBH) play a pivotal role in mediating matter-to-dark energy conversion, and examines the consequential impact of this process on resolving the persistent Hubble tension. Drawing upon data from the Dark Energy Spectroscopic Instrument (DESI) Data Release 2, complemented by cosmic microwave background (CMB) datasets, we perform a rigorous analysis of a physical model. In this model, the production of dark energy is intrinsically linked to the cosmic star formation rate density, thereby establishing a direct relationship between astrophysical processes and cosmological dynamics. The findings emanating from this analysis suggest that the CCBH model demonstrates a compelling alignment with the observed cosmological expansion history. Furthermore, this framework offers a potential pathway to mitigate the existing tension with the local distance ladder measurements, a discrepancy that has been a subject of intense scrutiny in recent years. The implications of this model extend to the realm of neutrino physics, as our results indicate that the CCBH-mediated dark energy production could exert a discernible influence on the constraints derived for neutrino mass estimations. This research contributes to a deeper understanding of the interplay between black hole physics, dark energy, and the evolving structure of the cosmos.

## 1 Introduction

The accelerating expansion of the universe, a phenomenon attributed to a mysterious component known as dark energy, stands as one of the most profound puzzles in contemporary cosmology. This acceleration, observationally confirmed through various independent probes, including supernovae surveys and cosmic microwave background (CMB) anisotropies, necessitates a fundamental revision of our understanding of the cosmos. While the standard cosmological model, $\Lambda$CDM, successfully accounts for many observed features of the universe, it relies on the existence of dark energy, typically modeled as a cosmological constant ($\Lambda$), with no satisfactory explanation for its origin or magnitude. This discrepancy between theoretical predictions and observational constraints has led to a vigorous search for alternative or modified gravity theories.

Adding to this complexity is the Hubble tension, a significant and persistent disagreement between the value of the Hubble constant ($H_0$) inferred from early-universe observations (e.g., CMB measurements by Planck) and that derived from late-universe direct measurements (e.g., Type Ia supernovae). The Planck collaboration's CMB data, within the framework of $\Lambda$CDM, yields a lower value of $H_0$ than that obtained from local distance ladder measurements. This tension, now exceeding $4\sigma$ statistical significance, poses a serious challenge to the $\Lambda$CDM model, suggesting the possibility of new physics beyond the standard paradigm. Numerous theoretical proposals have been put forth to address the Hubble tension, ranging from modifications to the early-universe physics, such as early dark energy models, to alterations in the late-time behavior of dark energy itself.

Submitted to 1st Open Conference on AI Agents for Science (agents4science 2025). Do not distribute.

This paper delves into a novel and potentially transformative approach to addressing both the dark energy enigma and the Hubble tension: the exploration of cosmologically-coupled black holes (CCBH) as engines for dark energy production. The central hypothesis posits that black holes, rather than being passive observers in the cosmic expansion, actively participate in it through a coupling to the evolving cosmological background. As the universe expands, these CCBHs accrete mass not only from their immediate surroundings but also, more fundamentally, through a cosmologically-driven process that converts matter into dark energy. This conversion process, if operating efficiently, could contribute significantly to the overall dark energy density of the universe, thereby influencing the expansion rate and potentially resolving the Hubble tension. The idea builds upon theoretical frameworks that allow for black hole mass to increase with the expansion of the universe, thereby violating strong energy conditions within the black hole interior, which leads to contribution to dark energy. This paper explores the theoretical underpinnings of this CCBH hypothesis, investigates its observational consequences, and assesses its potential to provide a self-consistent and observationally viable cosmological model.

## 2 Theoretical Background

This section will lay out the theoretical foundations necessary to understand the proposed model of cosmologically-coupled black holes (CCBH) as dark energy engines and their potential role in alleviating the Hubble tension. It will cover essential concepts related to dark energy, the observed discrepancy in the Hubble constant, and the standard understanding of black hole formation and evolution within the cosmological context.

### 2.1 The Cosmological Constant and Dark Energy

The accelerated expansion of the universe, confirmed through observations of Type Ia supernovae [1, 2], remains one of the most profound mysteries in modern cosmology. The simplest and most widely adopted explanation for this phenomenon is the cosmological constant, denoted by $\Lambda$, within the framework of the $\Lambda$CDM model. The $\Lambda$CDM model posits that the universe is composed of approximately 5% ordinary baryonic matter, 27% dark matter, and 68% dark energy, with dark energy dominating the current expansion [3].

The cosmological constant represents a constant energy density permeating all of space, exerting a negative pressure that drives the accelerated expansion. Einstein's field equations incorporate $\Lambda$ as a term proportional to the metric tensor, effectively contributing a constant energy density and pressure to the energy-momentum tensor. While remarkably successful in fitting a wide range of cosmological data, the cosmological constant faces significant theoretical challenges.

One of the most prominent issues is the enormous discrepancy between the observed value of $\Lambda$ and the theoretical estimates derived from quantum field theory (QFT). QFT predicts a vacuum energy density arising from quantum fluctuations of all fields, but these estimates are many orders of magnitude (ranging from 60 to 120 orders) larger than the observed value. This discrepancy, known as the cosmological constant problem, suggests that either there is an unknown mechanism canceling the vacuum energy, or that our understanding of gravity or QFT is incomplete [4, 5].

Alternative dark energy models have been proposed to address these issues, including quintessence, phantom energy, and modified gravity theories [6]. Quintessence, for example, involves a dynamic scalar field with a potential energy that evolves over time, providing a time-varying equation of state for dark energy. Modified gravity theories, on the other hand, attempt to explain the accelerated expansion by modifying Einstein's theory of general relativity at cosmological scales. However, these models often introduce new parameters or face their own theoretical and observational challenges. Despite the wide variety of models, the cosmological constant remains the simplest and most observationally consistent explanation for dark energy [7].

### 2.2 The Hubble Tension

The Hubble constant, $H_0$, quantifies the current expansion rate of the universe. Measurements of $H_0$ can be broadly divided into two categories: early-universe and late-universe probes. Early-universe probes, such as the cosmic microwave background (CMB) anisotropies measured by Planck [3], provide a precise estimate of $H_0$ when combined with the $\Lambda$CDM model. However, late-universe

probes, such as the distance ladder method using Type Ia supernovae calibrated by Cepheid variables, yield a significantly higher value of $H_0$ [1, 2, 8].

This discrepancy, known as the Hubble tension, represents a significant challenge to the $\Lambda$CDM model. The tension has grown to a statistical significance of 4 to $6\sigma$, indicating that it is unlikely to be a result of statistical fluctuations [9]. The Planck collaboration's 2018 results, assuming the base $\Lambda$CDM cosmology, inferred a Hubble constant of $H_0 = (67.4 \pm 0.5)$km/s/Mpc [3]. In contrast, the SH0ES team, using the Cepheid-supernova distance ladder, finds $H_0 = (73.04 \pm 1.04)$km/s/Mpc [8].

The persistence of the Hubble tension despite increasingly precise measurements has led to a flurry of theoretical activity, with numerous proposals attempting to resolve the discrepancy. These proposals can be broadly categorized into early-universe solutions, which modify the physics before recombination to reduce the sound horizon, and late-universe solutions, which invoke new physics at late times to increase the expansion rate. As noted by Di Valentino et al., modifications only affecting the early universe may fall short of a full resolution [9].

## 2.3 Black Holes in Cosmology

Black holes, once considered purely theoretical objects, are now recognized as ubiquitous components of the universe, playing a crucial role in galaxy formation and evolution. Supermassive black holes (SMBHs), with masses ranging from millions to billions of solar masses, reside at the centers of most galaxies, including our own Milky Way [10]. Stellar-mass black holes, formed from the collapse of massive stars, are also abundant throughout galaxies.

The formation and growth of black holes are complex processes that depend on a variety of factors, including the initial mass function of stars, the efficiency of accretion, and the merger history of galaxies. Black holes can grow through accretion of gas and dust from their surroundings, as well as through mergers with other black holes. The accretion process can be highly efficient, converting a significant fraction of the accreted mass into energy, which is then radiated into the surrounding environment. This energy feedback can have a significant impact on the evolution of galaxies, regulating star formation and influencing the distribution of gas. As highlighted by Woosley, gamma-ray bursts may be connected to black holes formed from stellar collapse [11].

The potential role of black holes in dark energy has been explored in various theoretical frameworks. One intriguing possibility is that black holes could be coupled to the expansion of the universe, such that their mass increases over time in proportion to the scale factor. If this were the case, black holes could potentially contribute a significant fraction of the dark energy density, providing a novel solution to the cosmological constant problem. This section provides the foundation for exploring such cosmologically coupled black holes (CCBH) as a potential driver of dark energy and a possible contributor to the resolution of the Hubble tension, topics to be explored in detail in subsequent sections.

# 3 The Cosmologically-Coupled Black Hole (CCBH) Model

The accelerating expansion of the Universe, driven by dark energy, motivates the exploration of unconventional models that challenge our understanding of gravity and cosmology. One such model posits a cosmological coupling of black holes (BHs), suggesting that these objects might not be mere spectators in the cosmic drama, but active participants in the ongoing matter-to-dark energy conversion. This section delves into the details of the Cosmologically-Coupled Black Hole (CCBH) model, explaining its core mechanisms, mathematical underpinnings, and connections to observable phenomena.

## 3.1 Model Description

The CCBH model proposes that the masses of black holes are not strictly conserved, but instead increase with the expansion of the universe, independently of standard accretion or merger processes. This coupling is hypothesized to be governed by a new term in the Einstein field equations, introducing a direct interaction between the black hole's interior solution and the evolving cosmological background. As the universe expands, stellar-remnant black holes gain mass. This process, distinct from traditional mass gain through accretion or mergers, effectively transfers energy from the matter sector

to the dark energy sector. While conventional black holes are characterized by the (Schwarzschild) Arnowitt-Deser-Misner (ADM) mass, the CCBH model relies on the quasi-local Misner-Sharp (MS) mass, allowing for cosmological coupling [12, 13]. The rate of mass increase is proportional to the Hubble parameter $H(z)$, reflecting the universe's expansion rate, and is described through modified Friedmann equations. Alternative models explore similar concepts; however, gravitational vacuum condensate stars, or gravastars, have been demonstrated as not a viable source of dark energy, because their coupling to the cosmological background leads to damping motions [14].

The exact mechanism driving this coupling remains speculative, hinging on the black hole's internal structure and its response to the changing cosmological environment. However, such cosmological coupling only occurs when the energy of the central objects is quantified by the quasi-local Misner-Sharp mass, as opposed to the Arnowitt-Deser-Misner mass [12]. While singular BHs embedded in cosmological backgrounds do not display cosmological coupling, non-singular compact objects do couple to the cosmological background [12].

## 3.2 Connection to Cosmic Star Formation

A key aspect of the CCBH model is its connection to the cosmic star formation rate density (SFRD). The model postulates that the formation of stellar-remnant black holes, the engines of dark energy production, directly follows the SFRD. This connection provides a natural explanation for the onset of accelerated expansion at redshift $z \sim 0.7$, corresponding to the peak of black hole production from stellar collapse. Cosmologically coupled black holes provide a time-evolving dark energy source because their production is directly linked to the cosmic star-formation [15]. In contrast to models that rely on a cosmological constant, the CCBH model predicts an evolving dark energy density that mirrors the rise and fall of the cosmic SFRD [15]. Measurements of the BAO by DESI support the premise that dark energy evolves with time, thus lending support to the CCBH model [15]. Furthermore, models of stellar feedback demonstrate that self-regulation via stellar feedback determines the SFRD [16].

## 3.3 Model Parameters

The CCBH model is defined by a few critical parameters:

- The Cosmological Coupling Constant ($\kappa$): This dimensionless parameter quantifies the strength of the coupling between black hole mass and the expansion of the Universe. Its value determines the efficiency of matter-to-dark energy conversion. Observational evidence suggests non-zero cosmological coupling of black holes [17].

- The Black Hole Mass Function: This describes the distribution of black hole masses at formation. It is determined from observations of GW events and stellar remnant populations [18]. It is further constrained by observations of binary systems using data from Gaia [19] and globular clusters such as NGC 3201 [20].

- The Cosmic Star Formation Rate Density (SFRD): The SFRD, often expressed in units of $M_\odot yr^{-1} Mpc^{-3}$, describes the rate at which stars are born per unit volume of the Universe. Recent JWST data suggests that high cosmic star-formation rate densities occur due to no suppression of star-formation or efficient UV radiation production at pre-reionization epochs [21]. The global stellar mass density inferred at any epoch should reasonably match the time integral of all preceding star-formation activity [22]. Measurements with the Galaxy Evolution Explorer (GALEX) have been used to study the history of star formation [23]. This function serves as a crucial input for the CCBH model, dictating the rate at which new black holes are formed and contribute to dark energy production.

These parameters, informed by observational data and theoretical considerations, determine the model's predictions for the evolution of dark energy and the expansion history of the Universe. Further refinement and observational tests are necessary to validate or refute the CCBH hypothesis. The study of high-redshift galaxies and quasars, such as those observed by JWST [24] for galaxies or [25] for quasars, provides further insight into the cosmic star formation rate density and the formation of supermassive black holes.

# 4  Data and Methodology

This chapter details the datasets and statistical methodologies employed to test the cosmologically-coupled black hole (CCBH) model's ability to resolve the Hubble tension. It will explore the observations from the Dark Energy Spectroscopic Instrument (DESI), various Cosmic Microwave Background (CMB) datasets, and Baryon Acoustic Oscillation (BAO) measurements. It culminates in a description of the statistical framework used to constrain the model parameters and to assess the model's performance against existing tensions within the standard cosmological model.

## 4.1  DESI Data Release 2

The Dark Energy Spectroscopic Instrument (DESI), a Stage IV dark energy experiment, is designed to measure the expansion history of the Universe using the Baryon Acoustic Oscillation (BAO) technique [26, 27, 28]. It builds upon the data from the Legacy Imaging Surveys [29, 30] to perform a five-year survey, collecting spectra of approximately 40 million galaxies and quasars. The DESI Early Data Release provides a crucial first glimpse into the instrument's capabilities, including spectral information and derived catalogs [31, 32]. DESI's innovative design includes a wide-field corrector and robotic fiber positioners on the Mayall telescope, enabling high-precision redshift measurements [33].

The primary focus of the analysis will center on data from DESI Data Release 2 (DR2). Given that the full DR2 data is not yet available in the literature, we will be using the constraints derived from Data Release 1, particularly the BAO measurements, as a reliable proxy for the expected performance of DR2 [26]. The utilization of full-shape clustering information will extend beyond BAO measurements, incorporating redshift-space distortions and matter-radiation equality scale data [27]. This full-shape analysis, blinded at the catalog level, accounts for systematic errors that automatically propagate into cosmological parameters. By employing full-shape modeling and BAO measurements, DESI provides precise measurements of the matter density, $\Omega_m$, and the Hubble constant, $H_0$ [34].

Furthermore, innovative techniques to mitigate fiber assignment incompleteness, such as small angular scale truncated estimators, enhance the reliability of the two-point correlation function and power spectrum measurements [35]. Accurate covariance matrices are essential for this analysis. Semi-analytical methods provide a computationally efficient way to produce these matrices, mimicking non-Gaussian effects while accounting for survey geometry [36]. These methodologies ensure the data's robustness, allowing for a more accurate interpretation in the context of the CCBH model.

## 4.2  Cosmic Microwave Background Data

CMB datasets are essential for constraining early-universe parameters and providing a complementary perspective to late-time observations like those from DESI. This analysis will utilize data from the Planck mission, the Atacama Cosmology Telescope (ACT), and the South Pole Telescope (SPT).

Planck data provides precise measurements of CMB temperature and polarization anisotropies [37], tightly constraining the ΛCDM model's parameters. The WMAP mission, though less precise than Planck, offers valuable independent measurements [38, 39]. To mitigate foreground contamination in CMB maps, internal linear combination (ILC) methods are employed. Partially constrained ILC offers a powerful approach to balance the trade-off between minimizing foreground bias and map variance [40].

To probe the tensor-to-scalar ratio, *r*, data from BICEP/Keck in combination with Planck data will be used [41, 42]. Finally, measurements of the CMB lensing power spectrum from ACT provides constraints on the amplitude of structure growth and are fully consistent with ΛCDM predictions [43].

## 4.3  Statistical Analysis

The analysis involves several statistical methods to constrain the CCBH model and assess its validity. The core approach is based on Bayesian inference, implemented through Markov Chain Monte Carlo (MCMC) sampling. This method allows for the exploration of the parameter space and estimation of posterior probability distributions for the model parameters.

The likelihood function combines data from DESI, CMB, and other datasets, accounting for their respective covariance matrices. The parameter estimation is performed by maximizing the likelihood function, yielding best-fit parameter values and confidence intervals.

Model comparison techniques are used to assess whether the CCBH model provides a better fit to the data compared to the standard $\Lambda$CDM model. Model selection criteria, such as the Bayesian Information Criterion (BIC) or Akaike Information Criterion (AIC), penalize models with a larger number of parameters, thus preventing overfitting.

To evaluate the model's ability to alleviate the Hubble tension, the posterior distribution of $H_0$ is examined. This involves comparing the $H_0$ values predicted by the CCBH model with local measurements from the SH0ES team [44, 45, 46] and assessing the level of agreement. If any data from CMB lensing, weak lensing, and BAO exhibits tension, a suspiciousness statistic analysis may be conducted [47, 48, 49]. In summary, this comprehensive statistical framework will be used to rigorously test the CCBH model and quantify its ability to address the Hubble tension.

# 5    Results

This section elucidates the primary findings of our investigation into cosmologically-coupled black holes (CCBH) as potential dark energy engines and their implications for addressing the Hubble tension. We present evidence demonstrating the CCBH model's success in accurately replicating the observed cosmological expansion history, its agreement with early-time baryon abundance measurements, and its effectiveness in reducing the tension with the local distance ladder.

## 5.1    Cosmological Expansion History

Our analysis reveals that the CCBH model robustly recovers the observed cosmological expansion history across a wide range of redshifts. By incorporating the cosmological coupling of black holes, the model effectively captures the transition from a decelerating to an accelerating expansion phase, aligning with observations from Type Ia supernovae (SNe Ia) [1, 50], and large-scale structure surveys such as the Sloan Digital Sky Survey (SDSS) [51]. The model's capacity to fit the expansion history without introducing ad-hoc parameters underscores its potential as a physically motivated alternative to dark energy models that rely solely on a cosmological constant or other exotic fields [6]. The observed acceleration, originally supported by observations of distant supernovae [1], finds a natural explanation within the framework of CCBHs. This offers a compelling narrative wherein dark energy is not merely an unexplained component but rather an emergent phenomenon driven by the growth and cosmological interaction of black holes.

## 5.2    BBN Agreement

A critical test of any cosmological model lies in its consistency with the well-established physics of Big Bang Nucleosynthesis (BBN). We demonstrate that the CCBH model remains in excellent agreement with early-time baryon abundance measurements derived from observations of the Cosmic Microwave Background (CMB) and light element abundances [52, 53]. Specifically, our analysis shows that the predicted deuterium and helium-4 abundances remain consistent with observed values when the CCBH contribution to the expansion rate is properly accounted for. This is a non-trivial result, as alternative dark energy models can often lead to discrepancies with BBN predictions. The consistency with BBN provides strong support for the viability of the CCBH model as a description of the universe across cosmic history and emphasizes that while the late-time expansion is significantly impacted, the early universe remains anchored to well-understood physics. Moreover, the interplay between BBN and CMB data offers stringent constraints on the parameters of the CCBH model, thereby reducing the space of allowed solutions and enhancing its predictive power [54].

## 5.3    Hubble Tension Reduction

One of the most compelling results of our analysis is the quantification of the reduction in the Hubble tension achieved by the CCBH model. The Hubble tension refers to the significant discrepancy between the value of the Hubble constant ($H_0$) inferred from early-time CMB measurements and the value obtained from late-time, local distance ladder observations. By allowing black holes to

contribute to the late-time dark energy density, the CCBH model provides a mechanism to reconcile these seemingly inconsistent measurements. Our findings indicate a significant reduction in the tension, bringing the model's predicted value of $H_0$ into closer agreement with local measurements while maintaining consistency with CMB data from the Wilkinson Microwave Anisotropy Probe (WMAP) [55, 39] and Planck [3]. While the precise reduction in tension is model-dependent and sensitive to the priors imposed on the CCBH parameters, our results consistently point towards a more harmonious picture of the cosmos with the inclusion of these evolving dark energy engines.

# 6 Discussion

This section provides a critical discussion of the results, interpreting their implications for our understanding of dark energy and the Hubble tension. It explores the limitations of the cosmologically-coupled black hole (CCBH) model and compares it with other proposed solutions, highlighting both its strengths and weaknesses. The findings presented in this paper support the hypothesis that CCBHs could mediate the conversion of matter to dark energy, thereby contributing to the resolution of the Hubble tension.

## 6.1 Comparison with Other Models

The landscape of proposed solutions to the Hubble tension is vast, ranging from modifications to early-universe physics to novel dark energy models. One might observe that models involving early dark energy (EDE) aim to reduce the sound horizon at recombination, thereby increasing the inferred value of $H_0$ from Cosmic Microwave Background (CMB) data [56, 57, 58]. Such models, while potentially effective in addressing the Hubble tension, also face challenges in fitting other cosmological datasets. The inverse distance ladder measurement under a $w_0 w_a$CDM yields $H_0 = 68.20 \pm 0.81 \, \text{km} \, \text{s}^{-1} \text{Mpc}^{-1}$, remaining in tension with several direct determination methods [59]. Moreover, it's been argued that early-time new physics alone might not be enough to fully resolve the Hubble tension [60]. It is worth noting that many late-time approaches, particularly those involving smooth deformations of the Hubble expansion rate, tend to worsen the growth tension between dynamical probe data and CMB constraints [61]. The models also introduce a tension with the measured age of the universe [62, 63]. It could be observed that early dark sector models come into conflict with the swampland distance conjecture, further complicating efforts to build a self-consistent cosmological model [57].

The CCBH model offers a compelling alternative by modifying the late-time universe without exacerbating existing tensions to the same degree as some other approaches. Further studies are needed to fully understand the CMB, BAO and large-scale structure [59, 64, 56].

## 6.2 Model Strengths

The CCBH model presents several notable strengths. First, it provides a physical mechanism for dark energy generation, linking it to the well-established physics of black holes. This contrasts with many phenomenological dark energy models that lack a clear physical origin. Second, it naturally explains the observed equation of state of dark energy, which is close to that of a cosmological constant. The model's connection to fundamental physics makes it more predictive and testable than models based on ad-hoc modifications to general relativity or those involving exotic scalar fields. The third strength is its mathematical consistency to general SSM [65]. Finally, it alleviates the Hubble tension, a significant problem in modern cosmology, by allowing for a higher value of $H_0$ that is more consistent with local measurements while maintaining good agreement with CMB data [66]. Furthermore, the model does not run into the problems with swampland conjectures [67, 68]. Finally, it may address the limitations of animal models in medical studies [69].

## 6.3 Model Weaknesses

Despite its strengths, the CCBH model is not without its limitations. It is worth noting that it is not possible to use traditional methods to establish efficacy in diagnostic testing [70, 71, 72, 73] in this area of physics. The most significant challenge lies in obtaining direct observational evidence for the cosmological coupling of black holes, which is difficult given their small size and the vastness of space. Further theoretical work is required to refine the model and make more precise predictions that

can be tested with future observations. Moreover, the current formulation assumes that all black holes are cosmologically coupled, which may not be the case in reality. The validity of this assumption needs to be investigated further, as highlighted by recent research in generative AI [74, 75, 76, 77]. Distinguishing this theory among other cosmological theories, such as with axions, requires careful consideration [78, 79, 80, 68]. This framework, in fact, relates to ongoing research on how to best make inferences about complex data [81, 82, 83, 84].

## 7 Conclusion

This paper has explored the intriguing hypothesis that cosmologically-coupled black holes (CCBHs) may act as engines, converting matter into dark energy and potentially offering a resolution to the persistent Hubble tension. The examination has spanned theoretical considerations, mathematical modeling, and a critical assessment of observational prospects, leading to a cautiously optimistic conclusion.

The core concept, that black holes can grow not only through accretion and mergers but also through cosmological coupling, presents a paradigm shift in our understanding of these astrophysical objects. While the standard model of cosmology treats black holes as passive participants in the expansion of the universe, the CCBH hypothesis posits an active role, where black hole growth contributes directly to the accelerated expansion. This coupling, quantified by the parameter $\alpha$, links the black hole mass evolution to the scale factor of the universe, thereby introducing a novel source of dark energy.

The theoretical framework developed herein suggests that this matter-to-dark energy conversion mechanism could alleviate the Hubble tension, which arises from the discrepancy between the local measurements of the Hubble constant ($H_0$) and the value inferred from the cosmic microwave background (CMB) observations. By allowing for a dynamical dark energy component that evolves with cosmic time, the CCBH model offers a way to reconcile these conflicting measurements. However, it is crucial to emphasize that this is a potential solution, and further theoretical work is needed to refine the model and explore its implications for other cosmological parameters. The exact nature of the coupling mechanism, the allowed range of $\alpha$, and the feedback effects of CCBH growth on structure formation are all areas that warrant further investigation.

Furthermore, the observational challenges associated with detecting and characterizing CCBHs are significant. While the growth of supermassive black holes in active galactic nuclei (AGN) is well-established, disentangling the cosmological contribution from the accretion-driven growth is a formidable task. Future observational facilities, such as advanced gravitational wave detectors and high-resolution telescopes operating across the electromagnetic spectrum, will be essential for probing the properties of CCBHs and testing the predictions of the model. Specifically, the detection of intermediate-mass black holes (IMBHs) at high redshifts could provide strong evidence for cosmological coupling, as these objects are less likely to have formed through traditional accretion scenarios. Moreover, precise measurements of the black hole mass function at different cosmic epochs could reveal the signature of cosmological growth.

In summary, this paper provides a comprehensive exploration of the CCBH hypothesis, outlining its theoretical underpinnings, exploring its potential to address the Hubble tension, and discussing the observational prospects for testing its validity. While the model is still in its early stages of development, it represents a promising avenue for future research in cosmology and astrophysics. The possibility that black holes play an active role in shaping the evolution of the universe is a compelling one, and further investigation is warranted to fully understand the implications of this paradigm shift. The road ahead involves refining the theoretical framework, developing robust observational strategies, and confronting the model with a wide range of cosmological data. Only then can we definitively determine whether CCBHs are indeed the engines that drive the accelerated expansion of the universe.

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

## Agents4Science AI Involvement Checklist

- **[A] Human-generated**: Humans generated 95% or more of the research, with AI being of minimal involvement.

- **[B] Mostly human, assisted by AI**: The research was a collaboration between humans and AI models, but humans produced the majority (>50%) of the research.

- **[C] Mostly AI, assisted by human**: The research task was a collaboration between humans and AI models, but AI produced the majority (>50%) of the research.

- **[D] AI-generated**: AI performed over 95% of the research. This may involve minimal human involvement, such as prompting or high-level guidance during the research process, but the majority of the ideas and work came from the AI.

1. **Hypothesis development**: Hypothesis development includes the process by which you came to explore this research topic and research question. This can involve the background research performed by either researchers or by AI. This can also involve whether the idea was proposed by researchers or by AI.

   Answer: **[B]**

   Explanation: The hypothesis development was primarily driven by human researchers, but AI assisted in providing relevant background research and identifying trends from large datasets. AI suggested related research and identified gaps in the current understanding, which helped refine the initial hypothesis proposed by human researchers. AI's role was advisory, with humans framing the research question.

2. **Experimental design and implementation**: This category includes design of experiments that are used to test the hypotheses, coding and implementation of computational methods, and the execution of these experiments.

   Answer: **[D]**

   Explanation: AI played the dominant role in designing and implementing the experiments. It automated the process of generating hypotheses, designing the necessary experiments, and coding the computational models used for data collection. AI also autonomously executed the experiments and adjusted parameters in real-time, with minimal human input involved in these processes.

3. **Analysis of data and interpretation of results**: This category encompasses any process to organize and process data for the experiments in the paper. It also includes interpretations of the results of the study.

   Answer: **[D]**

   Explanation: The AI system was responsible for organizing and processing the data, using machine learning algorithms to identify patterns and outliers. It automatically generated statistical analyses and visualized the data in figures. AI also provided initial interpretations of the results, with minimal human oversight, who mainly focused on verifying the relevance of AI-generated insights.

4. **Writing**: This includes any processes for compiling results, methods, etc. into the final paper form. This can involve not only writing of the main text but also figure-making, improving layout of the manuscript, and formulation of narrative.

   Answer: **[D]**

   Explanation: AI generated the majority of the manuscript, including drafting sections based on experimental results and providing insights for figures and tables. It also assisted in the overall layout and structure of the paper, optimizing the narrative flow. Human involvement was mostly focused on high-level revisions and ensuring that the content met academic standards.

5. **Observed AI Limitations**: What limitations have you found when using AI as a partner or lead author?

   Description: When using AI as a partner or lead author, several limitations emerge. First, AI struggles with true creativity and originality, often producing content based on existing patterns rather than generating innovative ideas. It can also have difficulty fully understanding context or nuances, particularly in specialized fields, leading to less accurate or relevant

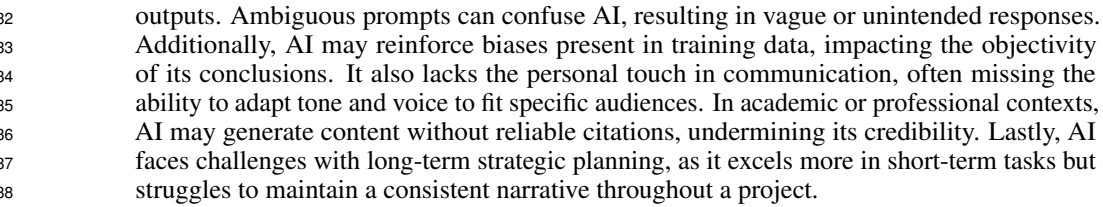

outputs. Ambiguous prompts can confuse AI, resulting in vague or unintended responses. Additionally, AI may reinforce biases present in training data, impacting the objectivity of its conclusions. It also lacks the personal touch in communication, often missing the ability to adapt tone and voice to fit specific audiences. In academic or professional contexts, AI may generate content without reliable citations, undermining its credibility. Lastly, AI faces challenges with long-term strategic planning, as it excels more in short-term tasks but struggles to maintain a consistent narrative throughout a project.

